# Pharmacological Potential and Mechanisms of Bisbenzylisoquinoline Alkaloids from Lotus Seed Embryos

**DOI:** 10.3390/biom15101377

**Published:** 2025-09-28

**Authors:** Yan Liu, Cong Wang, Qiong Liao, Canwei Du

**Affiliations:** 1Hunan Engineering Research Center of Lotus Deep Processing and Nutritional Health Sciences, School of Life and Health Sciences, Hunan University of Science and Technology, Xiangtan 411201, China; liuyan@mail.hnust.edu.cn (Y.L.); 2409010227@mail.hnust.edu.cn (C.W.); 2Hunan Provincial Key Laboratory of Animal Intestinal Function and Regulation, College of Life Sciences, Hunan Normal University, Changsha 410006, China

**Keywords:** lotus seed embryos, liensinine, isoliensinine, neferine, pharmacology, mechanism

## Abstract

Lotus seed embryos, a key component in traditional Chinese medicine, have attracted growing scientific interest due to their wide-ranging therapeutic potential. Among the bioactive compounds found in lotus seed embryos, three bisbenzylisoquinoline alkaloids—liensinine, isoliensinine, and neferine—stand out for their diverse pharmacological activities. These alkaloids are known to exhibit significant antitumor, anti-inflammatory, antihypertensive, neuroprotective, and antifibrotic effects, which make them promising candidates for the treatment of various chronic and acute diseases. Recent studies have highlighted their ability to modulate key signaling pathways involved in cancer progression, inflammation, fibrosis, and neurodegeneration. The precise mechanisms underlying their actions include modulation of oxidative stress, inhibition of pro-inflammatory cytokines, regulation of apoptosis, and modulation of cellular metabolism. This review aims to provide an in-depth overview of the pharmacological relevance of these alkaloids, focusing on their mechanisms of action and their therapeutic potential across different disease models. By synthesizing current evidence from preclinical studies, this review also lays a solid scientific foundation for future research, supporting the rational design and development of lotus-derived compounds for clinical application.

## 1. Introduction

*Nelumbo nucifera* Gaertn, commonly referred to as lotus, holy lotus, Indian lotus, or Chinese water lily [1], is a perennial aquatic herb of the family Nelumbonaceae. Characterized by its rhizomatous underground stems, peltate floating leaves, and distinctive nut-like fruits, the lotus exhibits exceptional ecological adaptability and multifunctionality. All anatomical structures of the plant—roots, stems, leaves, flowers, and seeds—possess significant economic, nutritional, and medicinal value. Based on morphological and agricultural characteristics, cultivated lotus varieties are classified into three major groups: root-stem lotus, cultivated for its thickened, edible rhizomes; seed lotus, selected for prolific seed yield and floral integrity; and flower lotus, prized for its ornamental diversity, with extensive variation in petal number, shape, and pigmentation [2]. Beyond its agricultural applications, the lotus has long been revered in Chinese traditional medicine for its wide-ranging pharmacological effects. Multiple studies have substantiated its efficacy in mitigating obesity, oxidative stress, inflammation, metabolic disorders, hepatic dysfunction, cardiovascular pathologies, malignancies, and neurodegenerative conditions, with notable antiviral, hypoglycemic, hypolipidemic, and memory-enhancing effects. Various clinical preparations—both single-herb and multi-component formulations—utilizing lotus extracts have been successfully implemented in integrative medicine [3].

Lotus seeds (Figure 1A), the edible and medicinally valuable reproductive structures of the plant, consist of four key anatomical components: the external seed coat, nutrient-rich cotyledons, internal seed coat, and the plumule (embryonic axis). Archaeological evidence indicates the use of lotus seeds for over 7000 years, reflecting their longstanding role as both food and therapeutic agents. Phytochemical investigations have revealed a complex matrix of bioactive constituents in lotus seeds, including flavonoids, glycosides, phenolic acids, and alkaloids, supporting their classification as functional foods with substantial pharmacological utility [4]. The lotus plumule (Figure 1B), embedded between the cotyledons, was first recorded in the *Compendium of Materia Medica*, noted for its bitter flavor and effects in clearing internal heat, alleviating psychological distress, calming mind, and promoting urination [5]. Chemically, the plumule contains a diverse array of chemical compounds, including alkaloids, flavonoids, organic acids, sterols, volatile oils, monosaccharides, water-soluble polysaccharides, and trace elements [6].

Among these chemical compounds, bisbenzylisoquinoline alkaloids are the most pharmacologically prominent, with approximately 20 distinct types identified to date that are widely distributed in roots, seeds, flowers and leaves. Liensinine, isoliensinine, and neferine are the most abundant alkaloids in the lotus seed embryo, each differing in concentration but sharing a closely related chemical structure [7] (Figure 1C). The crude extract of germinated lotus seed embryo was extracted and purified by ionic liquids pH-zone-refining countercurrent chromatography, with the gain of 37.3 mg liensinine, 57.7 mg isoliensinine and 179.9 mg neferine from 1.00 g germinated lotus seed embryo [8]. These three alkaloids have demonstrated therapeutic efficacy across a broad spectrum of pathological conditions, with significant potential in cancer treatment. In addition to their antitumor activity, these alkaloids exhibit diverse pharmacological properties, including antioxidant, astringent, emollient, diuretic, antidiabetic, antihyperlipidemic, antiaging, anti-ischemic, antiviral, anti-inflammatory, antiallergic, and hepatoprotective effects [9]. Their multifaceted actions across multiple physiological systems underscore their relevance as candidates for drug development.

This review aims to systematically summarize the latest research progress regarding these three major bisbenzylisoquinoline alkaloids in various diseases, with particular emphasis on molecular mechanism–based evidence. By offering a more targeted and practical reference, this work provided valuable insights for researchers and contributes to advancing the exploration of these natural compounds in drug development and clinical applications.

## 2. Antitumor Potential

Cancer arises from the disruption of normal regulatory mechanisms, leading to excessive proliferation and metastatic dissemination via the lymphatic and circulatory systems [10]. Conventional therapeutic modalities, including chemotherapy, radiotherapy, and targeted therapy, remain the primary interventions. However, these approaches are frequently associated with substantial financial burden and debilitating adverse effects, limiting their long-term tolerability and effectiveness [11]. Recent advances in biomedical research have facilitated the development of more refined anticancer strategies, including the incorporation of bioactive compounds derived from Chinese traditional medicine.

Bisbenzylisoquinoline alkaloids derived from the lotus plumule, including liensinine, isoliensinine, and neferine, have gained increasing attention for their potent anticancer activity (Figure 2). Increasing in vitro studies have demonstrated the antitumor mechanisms of liensinine, isoliensinine, and neferine across a wide spectrum of cancer cell lines, including colon (SW480), lung (PC9, A549), liver (HepG2, Huh-7), bladder (T24), gall bladder (GBC-SD, NOZ), prostate (LNCaP, DU-145, PC3), breast (MDA-MB-231, MCF-7), cervical (HeLa, SiHa), and esophageal (KYSE30, KYSE150, KYSE510) cancers, as well as non-tumorigenic mammary epithelial cells (MCF-10A), with multiple experimental approaches employed to assess cellular responses. For example, MTT and clonogenic survival assays have been used to quantify proliferation dynamics. Flow cytometric analysis has been performed to assess cell cycle distribution and apoptosis, while Hoechst 33342 staining has been applied to examine nuclear morphology. Mechanistic investigations have included Western blot analysis to measure the expression of regulatory proteins associated with cell cycle progression and apoptotic signaling pathways [12].

Liensinine has been shown to significantly suppress the proliferation of gastric cancer cells (BGC-823 and SGC-7901) when exposed to concentrations ranging from 20 to 120 μM. This suppression occurs primarily through the generation of reactive oxygen species (ROS) and the inhibition of PI3K/AKT signaling, leading to apoptosis and cell cycle arrest [13]. Liensinine at 80 μM disrupted redox homeostasis by increasing intracellular ROS, thereby impairing antioxidant defenses and inhibiting activation of the JAK2/STAT3 signaling pathway, promoting cell cycle arrest and apoptosis [14]. In gallbladder cancer models (GBC-SD and NOZ), liensinine induces G2/M arrest and apoptosis through inhibition of the zinc finger protein X-linked (ZFX)-induced PI3K/AKT pathway [12]. In non-small cell lung cancer (NSCLC), liensinine modulates autophagy by inducing autophagosome accumulation while simultaneously blocking autophagic flux by impairing lysosomal degradation capacity [15]. In hepatocellular carcinoma (HCC), target-based virtual screening revealed that liensinine interacted with the Kv10.1 potassium channel, inhibiting its currents in a concentration-dependent manner [16]. Further investigations in HUH7 and Hep1-6 cell lines demonstrated that liensinine inhibits cell viability, migration, and proliferation while promoting apoptosis [17]. In intrahepatic cholangiocarcinoma (ICC), in vitro assays confirmed its ability to suppress cell proliferation and inhibit epithelial–mesenchymal transition (EMT) through modulation of the HIF-1α-mediated TGF-β1/p-Smad3 signaling pathway [18]. In bladder cancer T24 cells, liensinine decreases the expression levels of p-AKT, CDK2, and CDK4 while increasing the expression of γH2AX, a marker of DNA damage and senescence [19]. In breast cancer, liensinine at 60 μM exhibits significant antitumor effects in MDA-MB-231 and MCF-7 cancer cells by increasing the Bax/Bcl-2 ratio, activating caspase-3, and inducing PARP cleavage, culminating in apoptosis, cell cycle arrest, and inhibition of migration and invasion [20]. Furthermore, liensinine at 20 μM binds to the S1 G-quadruplex structure in the fibroblast growth factor receptor 2 (FGFR2) promoter with a high affinity and selectivity, inhibiting FGFR2 expression at both the transcriptional and translational levels, providing a novel approach for breast cancer treatment [21]. In colorectal cancer, liensinine acts synergistically with oxaliplatin by inhibiting autophagy via suppressing HIF-1α/eNOS signaling, a key regulatory axis in oxaliplatin resistance [22].

Isoliensinine exhibits potent anticancer activity across various cancer types, with notable efficacy against triple-negative breast cancer. It triggers apoptotic cell death by inducing oxidative stress and activating both the p38 and JNK MAPK signaling pathways, showing selective cytotoxicity with minimal effects on normal human mammary epithelial cells (MCF-10A) [23]. In hepatocellular carcinoma, isoliensinine induces apoptosis in HepG2, Huh-7, and H22 cell lines by suppressing NF-κB signaling through promoting the dephosphorylation of the P65 subunit at Ser536 [24,25]. Additional studies demonstrate that 10 μM isoliensinine targets TGFBR1 and regulates the TGF-β-Smad signaling cascade to suppress the proliferation and migration of gastric cancer cell lines HGC27 and AGS [26]. Functional assays also identify isoliensinine as a novel AKT-binding ligand capable of suppressing the AKT/GSK-3α pathway and inducing cell cycle arrest and apoptosis in cervical cancer models [27]. Isoliensinine increases intracellular and mitochondrial ROS levels, disrupts mitochondrial membrane potential, impairs mitochondrial function, and suppresses cell viability in a dose-dependent manner at concentrations ranging from 0 to 40 μM [28]. Moreover, isoliensinine enhances the efficacy of conventional chemotherapeutics; co-administration with paclitaxel significantly amplifies cytotoxicity in multidrug-resistant HCT-15 colorectal cancer cells primarily through potentiation of mitochondria-mediated apoptosis [29].

Neferine exhibits broad-spectrum anticancer activity across multiple tumor types through diverse molecular mechanisms. In renal cancer models, neferine treatment induces apoptotic cell death through suppression of the NF-κB signaling pathway, mediated by caspase-dependent cleavage of the p65 (RelA) subunit [30]. In thyroid cancer, neferine demonstrates significant antitumor activity by inhibiting cell proliferation and promoting apoptosis via nuclear factor E2-associated factor 2 (Nrf2)/HO-1/NQO1 signaling modulation with 5 and 10 μM neferine [31]. It regulates p38 MAPK/JNK1/2 pathways to modulate melanoma proliferation, apoptosis, and oxidative stress [32], and inhibits TGF-β signaling to induce MST1/ROS-mediated pyroptosis in lung cancer cells [33]. In esophageal squamous cell carcinoma (ESCC), neferine exerts antiproliferative and pro-apoptotic effects by activating the ROS-dependent JNK signaling cascade, inducing G2/M phase arrest and apoptotic cell death [34]. In ovarian cancer cells, neferine selectively induces autophagy by activating the p38 MAPK/JNK pathway and inhibiting mTOR signaling, with minimal cytotoxicity in non-malignant oviduct epithelial cells [35]. In human prostate cancer cell lines (DU145 and LNCaP), neferine induces cell death by dual modulation of autophagic flux and JNK signaling pathway activation [36]. Subsequent investigations revealed its ability to induce ROS-mediated autophagy and apoptosis in androgen receptor (AR)-positive prostate cancer cells [37]. In cervical cancer cell lines (HeLa and SiHa), neferine upregulates key apoptotic proteins while downregulating antiapoptotic molecules such as Bcl-2, procaspase-3, procaspase-9, and translationally controlled tumor protein (TCTP) [38]. In MDA-MB-231 breast cancer cells, neferine modulates miR-374a and FGFR-2 expression, influencing the PI3K/AKT and MEK/ERK signaling pathways to inhibit proliferation [39]. Recent miRNA multi-omics analyses of endometrial cancer cells revealed that neferine activates mitochondrial apoptotic signaling through Ca^2+^-mediated endoplasmic reticulum stress and PI3K/AKT modulation, thereby promoting apoptosis [40]. Neferine has also been shown to potentiate the efficacy of doxorubicin (DOX) in human lung adenocarcinoma A549 cells through ROS-mediated apoptosis and MAPK activation, as well as potently inhibiting NF-κB nuclear translocation [41,42]. Additional studies on A549 cells have shown that neferine downregulates NF-κB and Bcl-2, upregulates Bax and cytochrome c, and activates caspase cascades, leading to apoptosis through DNA fragmentation. It also induces excessive ROS production, MAPK activation, lipid peroxidation, depletion of antioxidant reserves, mitochondrial membrane potential loss, intracellular calcium accumulation, and G1 phase cell cycle arrest in a dose-dependent manner [43]. It also potentiates cisplatin efficacy, lowering required doses and reducing toxicity [44].

Evidence from various in vivo animal models strongly supports the antitumor potential of these alkaloids, demonstrating their efficacy in suppressing tumor growth and metastasis with manageable toxicity. Liensinine has shown significant antitumor activity in several cancer models. In prostate cancer models, these alkaloids suppress 5-α-reductase activity and downregulate androgen receptor expression through modulation of the PI3K/AKT signaling pathway [45]. In hepatocellular carcinoma, it suppressed tumor growth in mouse models with efficacy comparable to cisplatin and oxaliplatin, while inducing fewer adverse effects [16]. Liensinine also enhanced the efficacy of radiotherapy and immunotherapy in subcutaneous xenograft and orthotopic liver cancer models. Similarly, in intrahepatic cholangiocarcinoma (ICC) models, liensinine significantly inhibited tumor growth in vivo [18]. Neferine has been extensively evaluated in a range of in vivo models. It has demonstrated antitumor activity in breast cancer models by inhibiting tumor growth and inducing apoptosis [35,46]. In pulmonary carcinogenesis models in Wistar rats, neferine effectively suppressed tumor development, restoring the expression of key proteins like p53, Bax, and caspase-3 [47]. Neferine has also been shown to enhance the efficacy of conventional chemotherapies. It potentiates the anticancer effects of cisplatin while reducing required dosages, thereby mitigating chemoresistance and toxicity [44].

In vitro evidence indicates that liensinine, isoliensinine and neferine inhibit multiple cancer cell lines by inducing ROS, cell-cycle arrest and mitochondrial/apoptotic pathways, and by modulating signaling cascades such as PI3K/AKT, MAPK/JNK and NF-κB. Distinctively, liensinine targets Kv10.1 channels and FGFR2 G-quadruplexes, isoliensinine directly binds AKT and modulates TGFBR1, while neferine uniquely reverses chemoresistance and induces pyroptosis. In vivo studies provide preliminary efficacy data and suggest synergy with radiotherapy, chemotherapy and immunotherapy. However, a translational gap remains—many in vitro effective concentrations are in the micromolar range while achievable systemic exposures following oral administration are likely lower. To support clinical translation, future work should prioritize comprehensive in vivo pharmacokinetics, standardized formulation development, and rigorous efficacy testing across multiple tumor models.

## 3. Neuroprotective Potential

Alzheimer’s disease (AD) is a multifactorial neurodegenerative disorder, with cholinergic and amyloid hypotheses [48]. Bisbenzylisoquinoline alkaloids from lotus seed embryos exhibit notable neuroprotective effects through modulation of oxidative stress, apoptosis, autophagy, and inflammatory pathways. Liensinine, isoliensinine, and neferine suppress tau hyperphosphorylation by inhibiting the Ca^2+^-CaM/CaMKII pathway, thereby reducing tau accumulation and protecting Aβ_25–35_-damaged PC12 cells [49]. In vitro studies with APP695swe SH-SY5Y cells and in vivo transgenic *C. elegans* confirm that liensinine and neferine enhance cell viability, inhibit Aβ accumulation, reduce ROS, and activate autophagy to mitigate Aβ/tau toxicity [50]. Moreover, Tianjin Anti-cancer Special Extract of Nelumbo nucifera (TASENN), a formulation enriched in these alkaloids, reduces tau phosphorylation and neurofibrillary tangle (NFT) formation in APP/PS1 mice and protects PC12 cells from Aβ damage [51].

Liensinine, neferine, and isoliensinine exhibit strong antineuroinflammatory effects in lipopolysaccharide (LPS)-activated microglia by reducing nitric oxide (NO) and pro-inflammatory cytokines, including tumor necrosis factor-alpha (TNF-α), interleukin-1 beta (IL-1β), and interleukin-6 (IL-6). They inhibit IκBα phosphorylation/degradation through suppression of ROS generation and free radical scavenging [52]. Lotus plumule alkaloids have also demonstrated neuroprotective effects across various other neurological disease models, including Parkinson’s disease (PD), ischemic brain injury, hypoxic–ischemic encephalopathy, epilepsy, and depression. In PD models involving LPS-activated microglial cultures and MPTP-treated mice, neferine has been shown to markedly suppress neuroinflammation by inhibiting the NF-κB signaling pathway. This effect is mediated through the prevention of IκBα phosphorylation and p65 nuclear translocation, resulting in attenuated pro-inflammatory cytokine production [53].

Liensinine has also been investigated in models of ischemia–reperfusion (I/R) injury. Pretreatment with liensinine or rapamycin (RAPA) in an external I/R neuronal model has been shown to increase cell viability, reduce cellular damage and apoptosis, and inhibit autophagy. Liensinine exerts its neuroprotective efficacy through positive regulation of the PI3K/Akt signaling pathway [54].

In rats with permanent brain occlusion, neferine reduces neurological deficits, infarct volume, oxidative stress, and apoptosis, accompanied by lowering 4-hydroxynonenal (4-HNE), NO, neuronal nitric oxide (nNOS), calcium, Bax, caspase-3 and increased Hsp70, Bcl-2 expression [55]. Neferine also protects against hypoxic–ischemic encephalopathy (HIE) by suppressing neuronal pyroptosis. It downregulates caspase-1, ASC, and GSDMD, reduces IL-18 and IL-1β, and alleviates neuroinflammation and oxidative stress. In CoCl_2_-induced HIE models, neferine further inhibits nod-like receptor family pyrin domain (NLRP3) inflammasome–mediated pyroptosis, highlighting its therapeutic potential against neonatal brain injury [56]. In the context of excitotoxicity, neferine has been shown to inhibit glutamate release induced by the potassium channel blocker 4-aminopyridine in a dose-dependent manner. In addition, neferine activates 5-Hydroxytryptamine (5-HT1A) receptors in cortical synaptosomes, thereby reducing calcium influx and glutamate release via Gi/o protein activation and inhibition of the adenylate cyclase (AC)/cAMP/protein kinase A (PKA) cascade, thereby contributing to neuroprotection [57]. In kainic acid–induced seizures, neferine lowers NLRP3, caspase-1, IL-18, IL-1β, IL-6, TNF-α, while enhancing glutamate and synaptic protein expression, reducing glial activation and neuronal injury, indicating its potential as a treatment for epilepsy [58]. In ischemic stroke models, it protects PC12 cells from oxidative stress, preserves mitochondrial function, boosts adenosine triphosphate (ATP), and activates Nrf2 signaling; in vivo, it improves neural scores, infarct volume, cerebral blood flow, and antioxidant enzyme function [59]. Additionally, in chronic stress–induced depression, neferine enhances monoamine neurotransmitter secretion and modulates gut microbiota, particularly *Lactobacillus*, suggesting antidepressant potential [60].

Preclinical in vitro and in vivo studies indicate that liensinine, isoliensinine and neferine exert neuroprotective effects by suppressing neuroinflammation, mitigating oxidative stress, reducing tau hyperphosphorylation and modulating autophagy/mitochondrial homeostasis in models of Alzheimer’s disease, Parkinson’s disease and ischemic injury (Figure 3). While mechanistic data are promising, further dose–response, long-term safety, behavioral outcome and multi-species in vivo studies are needed to define translational potential.

## 4. Anti-Inflammatory Potential

Bisbenzylisoquinoline alkaloids have demonstrated significant anti-inflammatory activity across a range of models (Table 1). Liensinine modulates macrophage ferroptosis and ameliorates acute liver injury by regulating iron metabolism and autophagy, specifically inhibiting ferritinophagy-mediated iron release through blocking autophagosome-lysosome fusion [61]. In LPS-induced acute lung injury (ALI), it acts through PI3K/AKT/mTOR signaling to reduce inflammatory responses [62]. Additionally, liensinine alleviates oxidative stress and inhibits NF-κB phosphorylation and NLRP3 inflammasome synthesis, reducing pro-inflammatory cytokine release in sepsis models [63]. Liensinine also suppresses IL-1β-induced chondrocyte inflammation via NF-κB inhibition and improves cartilage damage in osteoarthritis models [64], indicating its potential utility in osteoarthritis treatment.

Neferine has also been shown to exert anti-inflammatory effects. In vitro experiments using TNF-α/IFN-γ-stimulated human keratinocytes (HaCaT) have demonstrated that neferine suppresses phosphorylation of p38, JNK, and ERK. In vivo, neferine treatment substantially alleviates 2,4-dinitrochlorobenzene (DNCB)-induced atopic dermatitis-like symptoms in mice, supporting its medical potential for the treatment of inflammatory skin conditions [65]. Neferine has also been shown to attenuate vascular inflammation by downregulating the mRNA and protein expression levels of intercellular adhesion molecule 1 (ICAM1) and vascular cell adhesion molecule 1 (VCAM1) via the inhibition of NF-κB signaling [66]. In lysophosphatidylcholine (LPC)-stimulated human umbilical vein endothelial cells (HUVECs), neferine exhibits endothelial-protective effects byregulating the dimethylarginine dimethylaminohydrolase (DDAH)-asymmetric dimethylarginine (ADMA)-NO pathway. Mechanistic studies have reported that neferine preserves DDAH enzymatic activity, reduces pathological ADMA accumulation, and lowers oxidative stress markers such as ROS and malondialdehyde (MDA), thereby restoring NO bioavailability and attenuating LPC-induced endothelial dysfunction [67].

Isoliensinine demonstrates anti-inflammatory and chondroprotective effects in osteoarthritis by reducing extracellular matrix degradation, NLRP3 activation, and Matrix metalloproteinase 3 (MMP-3) expression while inhibiting MAPK/NF-κB signaling and chondrocyte pyroptosis [68].

Both in vitro and in vivo studies demonstrate that liensinine, isoliensinine and neferine attenuate inflammatory responses by inhibiting NF-κB, MAPK and NLRP3 pathways and by modulating ferritinophagy/autophagy, leading to reduced pro-inflammatory cytokine production and tissue protection. Heterogeneity among models and dosing regimens highlights the need for standardized comparative studies to define optimal indications and dosing.

## 5. Anti-Hypertensive Potential

Bisbenzylisoquinoline alkaloids have demonstrated significant blood pressure-lowering effects (Table 2).

Neferine has been shown to effectively improve vascular remodeling in spontaneously hypertensive rats by inhibiting abnormal activation of the PI3K/AKT and TGF-β1/Smad2/3 signaling pathways [69]. In hypoxia induced H9c2 cells, neferine treatment significantly reduces cardiomyocyte apoptosis, restores mitochondrial membrane potential, and decreases ROS accumulation. Similar outcomes have been observed in spontaneously hypertensive rats treated with neferine at various concentrations in vivo, confirming its ability to effectively mitigate apoptosis in hypertensive cardiomyocytes [70].

Isoliensinine dose-dependently inhibits angiotensin II (Ang II)-induced proliferation of porcine coronary artery smooth muscle cells (CASMC) by suppressing platelet-derived growth factor-β (PDGF-β), basic fibroblast growth factor (bFGF), and proto-oncogenes (c-Fos, c-Myc), and modulating heat shock protein 70 (HSP70) expression, indicating its multi-target action against Ang II-activated vascular remodeling [71]. Liensinine exhibits hypotensive effects in Ang II-induced hypertension by reducing abdominal aortic wall thickening, limiting collagen deposition in ventral aortic vessels, attenuating proliferative markers (PCNA) expression, and suppressing the reduction of α-smooth muscle actin (α-SMA). It also blocks activation of MAPK/TGF-β1/Smad 2/3 signaling, thereby suppressing Ang II-induced vascular remodeling in hypertensive models [72].

These compounds have shown therapeutic potential in pulmonary hypertension, as evidenced by network pharmacology and experimental studies. Notably, studies have reported therapeutic effects on pulmonary hemodynamics and cardiac remodeling, with the three alkaloids significantly reducing right ventricular systolic pressure (RVSP), inhibiting maladaptive right ventricular hypertrophy, restoring extracellular matrix homeostasis, and suppressing proliferative signaling in pulmonary hypertension [73]. Preclinical data suggest that liensinine, isoliensinine and neferine ameliorate hypertension and vascular remodeling by inhibiting VSMC proliferation and modulating MAPK/TGF-β/PI3K-AKT signaling, thereby improving hemodynamics and structural remodeling in animal models. Further long-term in vivo studies with hemodynamic endpoints and safety profiling are required to determine clinical relevance.

## 6. Anti-Fibrotic Potential

Multiple studies have demonstrated the significant antifibrotic potential of liensinine, isoliensinine, and neferine, highlighting their ability to modulate key fibrotic pathways and reduce fibrosis in various tissues. These alkaloids have been shown to effectively inhibit the progression of fibrosis in experimental models, suggesting their promising therapeutic value for fibrotic diseases (Figure 4).

In lung fibroblasts, all three alkaloids have been shown to inhibit TGF-β-induced proliferation in both normal cells and fibroblasts derived from pulmonary fibrosis, significantly downregulating α-SMA protein expression and suppressing fibroblast migration capacity in both cell types [74].

Neferine demonstrates broad antifibrotic activity. In pulmonary fibrosis models, it restores superoxide dismutase (SOD) activity, reduces MDA and myeloperoxidase (MPO) levels, and decreases pro-inflammatory cytokines (TNF-α, IL-6, endothelin-1). It also suppresses NF-κB activation and TGF-β1 upregulation in macrophages [75]. Additionally, neferine alleviates doxorubicin-induced cardiotoxicity by enhancing Sirtuin 1 expression and activity, modulating TGF-β1 and p53, thereby reducing fibrosis, hypertrophy, and apoptosis in cardiomyocytes [76]. In atrial fibrosis, it reverses mitochondrial depolarization, lowers ROS, and boosts GSH and SOD activity—inhibiting atrial fibrillation and fibrosis via Nrf2/HO-1 activation and TGF-β/p-Smad2/3 suppression [77]. Neferine also attenuates endometrial fibrosis by inhibiting fibronectin, collagen I, CTGF, and α-SMA expression through TGF-β/ERK modulation [78]. In diabetic cardiomyopathy, it reduces collagen overproduction and ECM deposition in cardiac fibroblasts via coordinated inhibition of TGF-β1/Smad, ERK, and p38 MAPK pathways [79].

Isoliensinine also exerts antifibrotic effects. Notably, in spontaneously hypertensive rats, isoliensinine significantly alleviates renal injury and collagen accumulation by inhibiting TGF-β1/Smad2/3 signaling, leading to improved renal structure and function [80].

Liensinine, isoliensinine and neferine demonstrate antifibrotic activity across pulmonary, cardiac, endometrial and diabetic cardiomyopathy models, primarily via suppression of TGF-β/Smad, ERK/MAPK signaling and oxidative stress. Additional organ-specific dosing and long-term efficacy studies are warranted to validate therapeutic potential.

## 7. Antiarrhythmic Potential

Extensive preclinical studies have demonstrated that the three principal alkaloids of lotus plumule exhibit potent antiarrhythmic activity.

Neferine exerts concentration-dependent electrophysiological effects on cardiac tissue. In guinea pig ventricular myocardium, it prolongs action potential duration (APD), reduces the maximum upstroke velocity (Vmax), and diminishes the amplitude of slow response action potentials. It also suppresses ouabain-induced afterdepolarizations and decreases contractility. These effects result from concurrent inhibition of cardiac INa, IK, and ICa currents, which collectively stabilize myocardial electrical activity [81]. Neferine further blocks Kv4.3 channels in both open and inactivated states, contributing to its antiarrhythmatic properties in rabbit hearts [82]. In human embryonic kidney 293 (HEK 293) cells expressing Nav1.5 channels, neferine delays activation time to peak, prolongs inactivation, and slows recovery from inactivation, confirming Nav1.5 inhibition as a key antiarrhythmic mechanism [83]. It also enhances the pharmacokinetics and efficacy of amiodarone against both supraventricular and ventricular arrhythmias [84]. Additionally, neferine inhibits human ether-à-go-go-related gene (HERG) K^+^ channels by altering their gating kinetics, further supporting its antiarrhythmic role [85].

Liensinine demonstrates concentration-dependent (1–100 μM) antiarrhythmic effects, reducing systolic force in isolated left atrium and decreasing spontaneous beating rate in right atria [86]. Both liensinine and neferine suppress ventricular arrhythmias, though neferine is more potent at concentrations below 10 μM. This difference is attributed to neferine’s ability to bind both open and inactivated states of HERG channels, whereas liensinine only binds the open state [87].

Isoliensinine also exhibits antiarrhythmic activity by effectively suppressing early (EADs) and delayed afterdepolarizations (DADs) in rabbit ventricular myocytes. It achieves this through inhibition of INaL and ICaL currents, thereby stabilizing ventricular electrical activity [88].

Electrophysiological study indicated that liensinine, isoliensinine and neferine modulate multiple cardiac ion channels, prolong action potential duration and suppress EAD/DAD, suggesting antiarrhythmic potential. Given the dose-dependent channel effects and proarrhythmic risk, comprehensive safety and drug-interaction studies in larger animal models are necessary.

## 8. Antioxidant Potential

Lotus plumule alkaloids demonstrate clinically relevant antioxidant properties (Figure 5).

Neferine reduces oxidative stress and inflammation in CCl_4_-induced liver fibrosis by inhibiting MAPK and NF-κB/IκBα pathways. It enhances antioxidant enzyme such as SOD, GSH peroxidase (GPx), and catalase (CAT), while reducing MDA levels [89]. In LPS-induced septic myocardial injury, neferine improves cell viability and mitochondrial function in H9c2 cardiomyocytes via PI3K/AKT/mTOR activation, reducing apoptosis and ROS production [90]. Under hypoxia/reoxygenation stress, it activates SIRT1/Nrf2 signaling, upregulates heme oxygenase-1, and alleviates mitochondrial dysfunction [91]. Neferine also demonstrates neuroprotection through NF-κB suppression, inhibition of lipid peroxidation and protein nitration, modulation of NO homeostasis, and dual inhibition of acetylcholinesterase (AChE) and β-site amyloid precursor protein cleaving enzyme 1 (BACE1) [92]. In aluminum chloride-induced AD mice, oral administration of neferine has been shown to inhibit ROS production, increase antioxidant SOD and CAT expression, and reduce GSH levels, collectively demonstrating its significant antioxidant pharmacotherapeutic potential [93]. In UV-induced skin photoaging, neferine effectively delays wrinkle formation by enhancing SOD and GPx activities, reducing oxidative stress, inhibiting epidermal hypertrophy, and preserving collagen integrity [94].

Isoliensinine protects HT-22 hippocampal neurons from glutamate-induced ferroptosis via Nrf2/GPX4 activation. It reduces mitochondrial depolarization, iron overload, and accumulation of MDA and ROS, while maintaining GPx, SOD, solute carrier family 7 member 11 (SLC7A11), and GPx4 levels and suppressing acyl-CoA synthetase long-chain family member 4 (ACSL4). Its radical-scavenging and iron-chelating capacities are comparable to those of classic ferroptosis inhibitors [95].

Liensinine demonstrates potent hepatoprotective activity against septic liver injury by activating the Nrf2 pathway, ameliorating lipid peroxidation, and enhancing antioxidant enzyme function. It significantly mitigates markers of cellular damage and strengthens hepatic antioxidant defense mechanisms, supporting its potential as a therapeutic candidate for sepsis-associated liver dysfunction [96].

All three alkaloids show significant inhibitory effects on t-BHP-induced oxidative stress and cytotoxicity in HepG2 cells. Neferine exhibits the strongest protective effect by inhibiting ROS and thiobarbituric acid reactive substance (TBARS) formation, reducing lactate dehydrogenase (LDH) release, and increasing GSH levels. Overall, these compounds provide substantial protection against oxidative damage in vitro [97]. Liensinine, isoliensinine and neferine enhance endogenous antioxidant defenses and suppress oxidative markers across hepatic, cardiac, and neural models, indicating broad cytoprotective effects. Translational studies should define dosing and long-term outcomes in chronic disease models.

## 9. Antidiabetic Potential

Neferine enhances glucose metabolism in L6 myoblasts by promoting glucose transporter 4 (GLUT4) expression and translocation to the plasma membrane, thereby increasing membrane fusion and glucose uptake. These effects are mediated through activation of the G protein-PLC-PKC and adenosine monophosphate (AMP)-activated protein kinase (AMPK) pathways, which upregulate GLUT4 and facilitate its membrane fusion. Neferine also elevates intracellular Ca^2+^ via the G protein-PLC-IP3-IP3R axis, further stimulating glucose uptake [98]. In models of type 2 diabetes, neferine regulates chemokine signaling in the superior cervical ganglion (SCG), downregulating both chemokine ligand 5 (CCL5) and its receptor (CCR5) at the transcriptional level. This suppresses aberrant neuronal signaling within the SCG and ameliorates cardiovascular autonomic neuropathy, suggesting a therapeutic role in diabetic neuroinflammation [99]. Additionally, neferine reduces ROS, normalizes SOD and MDA levels, and inhibits PI3K/Akt and NF-κB pathways in vascular models, thereby protecting endothelial cells from apoptosis and suggesting benefits in diabetes-induced vascular dysfunction [100].

Isoliensinine also demonstrates antidiabetic and antidyslipidemic activity in both in vitro and in vivo models. In L6 cells, isoliensinine administration has been shown to increase GLUT4 translocation by 2.5-fold. In KK-Ay diabetic mice, treatment exerts significant positive effects on serum insulin levels, fasting blood glucose, and body weight, as well as on GLUT4 protein and phosphorylated AMPK levels [101].

Preclinical data indicate that neferine and isoliensinine improve glucose metabolism by promoting GLUT4 translocation, activating AMPK and modulating autonomic/chemokine signaling (Figure 6). Chronic diabetic animal studies are needed to confirm metabolic outcomes, safety and interactions with standard antidiabetic agents.

## 10. Other Potential

Liensinine demonstrates therapeutic potential in non-alcoholic fatty liver disease (NAFLD) by improving metabolic disorders, insulin resistance, and dyslipidemia in high-fat diet (HFD)-fed mice. These benefits are mediated through TAK1/AMPK signaling activation, which reduces lipid accumulation in palmitic acid-treated cells and suppresses excessive ROS generation [102]. Additionally, liensinine attenuates ischemic myocardial injury and DNA damage by inhibiting Wnt/β-catenin signaling [103]. It also acts as a novel inhibitor of myostatin (MSTN), disrupting its binding to activin receptor IIB (ActRIIB) and downregulating atrophy-related proteins muscle RING-finger protein-1 (MuRF-1) and muscle atrophy F-box (MAFbx)/Atrogin-1, thereby promoting skeletal muscle regeneration [104].

Isoliensinine effectively inhibits osteoclast differentiation and bone resorption by competitively blocking receptor activator of nuclear factor κB ligand (RANKL)-RANK interaction, leading to suppression of MAPK (p38, JNK, ERK) and NF-κB pathways. It also reduces ROS accumulation and stabilizes Ca^2+^ flux in osteoclast precursors [105]. In ovariectomized (OVX) mice, isoliensinine alleviates bone loss by inhibiting NF-κB signaling and improving bone microarchitectural parameters such as trabecular thickness, number, and bone volume fraction [106].

Neferine exhibits broad protective effects across multiple disease models. In diabetic nephropathy, it reduces blood glucose, creatinine, urea, and miR-17-5p levels, activates Nrf2, enhances antioxidant activities (SOD, GPx), and ameliorates renal fibrosis [107]. In ovalbumin (OVA)-induced asthma models, neferine exerts immunomodulatory effects at doses of 20 mg/kg and 40 mg/kg by reducing systemic and pulmonary inflammation. It decreases Th2 cytokines, IgE, and pro-inflammatory mediators, improves airway resistance, and suppresses pulmonary inflammation and collagen deposition via MAPK inhibition [108]. Neferine also exhibits potent antiplatelet and antithrombotic activities. In murine models, it dose-dependently inhibits platelet activation, adhesion, and aggregation induced by collagen, thrombin, U46619 (washed platelets), and ADP (platelet-rich plasma). At 6 mg/kg, neferine significantly reduces thrombus formation in vivo and uniquely promotes disintegration of preformed platelet aggregates, a property not typically observed with conventional antiplatelet agents [109]. During adipogenesis in 3T3-L1 cells, neferine has been shown to inhibit lipid accumulation in a dose-dependent manner. Neferine increases AMPK activity, reduces fat production, promotes lipid metabolism, and enhances acetyl-CoA carboxylase (ACC) phosphorylation, thereby promoting fatty acid oxidation [110]. Furthermore, it mitigates aging-related liver dysfunction by inducing autophagy via DAPK1 and JNK signaling, restoring mitochondrial function, and reducing senescence markers [111].

Beyond the primary pharmacological activities, the alkaloids show effects on bone remodeling, platelet function, anti-aging and renal protection, indicating multi-target potential. Most evidence is from single short-term models, systematic and reproducible in vivo studies are required to evaluate feasibility and risk-benefit.

## 11. Discussion

Lotus plumule, a key botanical in traditional Chinese medicine, exhibits considerable therapeutic potential due to its rich alkaloid content. These bioactive alkaloids display a broad spectrum of pharmacological activities, encompassing anticancer, neuroprotective, anti-inflammatory, antifibrotic, antihypertensive, antiarrhythmic, antioxidant, and antidiabetic effects. Their low toxicity and multifaceted efficacy establish lotus plumule as a promising candidate in integrative medicine, offering a natural therapeutic alternative with reduced adverse effects compared to synthetic agents.

Mechanistically, these alkaloids act on target cells by modulating key signaling pathways such as PI3K/AKT, NF-κB, TGF-β, and MAPK/JNK. They also induce autophagy, including mitophagy and ferritinophagy, suppressing the proliferation of pathological cells. These bioactive compounds regulate programmed cell death pathways, control cell cycle checkpoints, and dynamically influence intracellular signaling, collectively contributing to their therapeutic effects.

Growing global interest in preventive healthcare has led to increased recognition of the therapeutic potential of natural botanicals. Among these, lotus plumule has emerged as one of the prominent medicinal agents for pharmacological investigation. Preclinical evidence from in vivo and in vitro studies supports the therapeutic promise of its bioactive alkaloids. However, challenges remain regarding clinical translation. While the pre-clinical data summarized herein demonstrates the remarkable pharmacological potential of lotus plumule alkaloids, a critical translational challenge must be acknowledged. The effective concentrations (IC_50_/EC_50_) in most in vitro studies are in the micromolar range, whereas the achievable plasma concentrations following oral ingestion of a traditional lotus plumule tea infusion are expected to be orders of magnitude lower, likely in the nanomolar range, due to limited solubility and poor bioavailability inherent to many bisbenzylisoquinoline alkaloids [112,113]. Pharmacokinetic studies on neferine support this notion, showing low oral bioavailability and rapid metabolism [114]. Consequently, while occasional consumption of lotus plumule tea may offer mild health benefits, potentially attributable to synergistic effects of its various constituents, it is unlikely to produce the robust therapeutic effects observed in laboratory settings. At the same time lotus-plumule products also contain higenamine, a plant-derived β2-agonist that is prohibited by the World Anti-Doping Agency (WADA) at all times. In a human study using commercially available lotus-plumule products, repeated intake equivalent to ~680 µg higenamine per dose led all participants to exceed the WADA reporting threshold of 10 ng/mL for free higenamine in urine, underscoring a tangible risk of adverse analytical findings (AAFs) in doping control. Given the batch-to-batch variability reported for plant-derived products and the possibility of undeclared higenamine, not only should athletes and individuals subject to anti-doping rules avoid lotus-plumule–containing products, but the general population should also use such products prudently and only under appropriate guidance [115,116]. To harness the full potential of these compounds for modern therapeutics, future research must prioritize the development of standardized, concentrated extracts and innovative formulations.

Beyond pharmacokinetic limitations, current research on lotus plumule alkaloids is constrained by an incomplete mechanistic understanding. Studies have predominantly focused on the PI3K/AKT axis, while other potentially relevant signaling pathways remain underexplored. This narrow scope hinders comprehensive insight into the therapeutic actions of the compounds. Furthermore, the pharmacological activity of lotus plumule alkaloids has primarily been evaluated in cellular systems and small animal models, with limited progression to clinical studies [117]. A substantial evidence gap remains regarding their efficacy and mechanisms in human subjects. Continued research, particularly in vivo and clinical investigations, is essential to elucidate the pharmacodynamics of individual alkaloids and establish their applicability in disease prevention and treatment.

In addition to the pharmacological insights, the therapeutic development of bisbenzylisoquinoline alkaloids from lotus seed embryos requires a structured translational framework. Firstly, mechanistic validation across multiple disease models is essential to confirm the consistency of signaling pathways such as PI3K/AKT, MAPK, NF-κB, and TGF-β/Smad. Secondly, dose–response and pharmacokinetic/pharmacodynamic profiling should be performed to determine therapeutic windows and optimize efficacy–toxicity balance. Thirdly, formulation strategies aimed at enhancing solubility, stability, and oral bioavailability are critical to address the limitations of these natural alkaloids. Fourthly, systematic safety and toxicity studies in animal models will provide indispensable data for translational evaluation. Finally, early-phase clinical trials guided by well-defined biomarkers of efficacy and safety represent the most appropriate step toward establishing therapeutic regimens. This staged approach highlights both the translational promise and the rigorous validation required before clinical application.

## Figures and Tables

**Figure 1 biomolecules-15-01377-f001:**
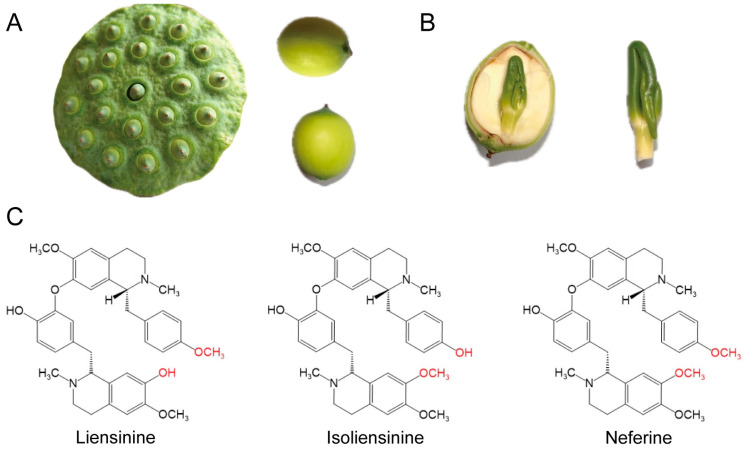
Morphology, anatomy, and chemical structures of lotus seed embryos and their alkaloids. (**A**) Morphology of lotus seeds and seed pods, showing the external structure of lotus and the embedded seeds. (**B**) Anatomical structure of lotus plumule, showing the internal germ and its distinct green plumule part. (**C**) Chemical structures of liensinine, isoliensinine, and neferine. They share a bisbenzylisoquinoline skeleton but differ in the substitution of different groups marked in red in the figure.

**Figure 2 biomolecules-15-01377-f002:**
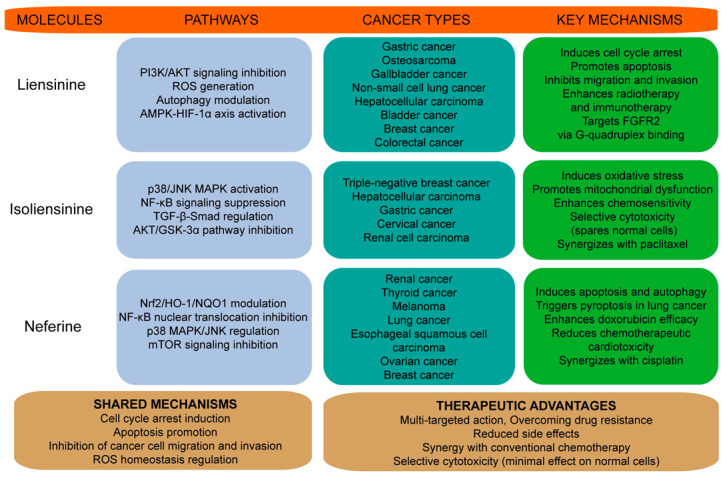
Anticancer mechanisms of liensinine, isoliensinine, and neferine across various tumor types. This figure illustrates how liensinine, isoliensinine, and neferine induce apoptosis, autophagy, and regulate the cell cycle across different cancer types. It highlights their key mechanisms of action and common pathways, along with their potential therapeutic value in cancer treatment.

**Figure 3 biomolecules-15-01377-f003:**
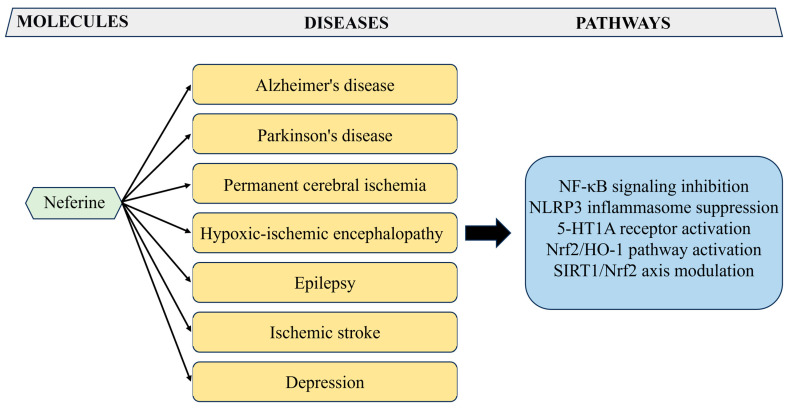
Neuroprotective potential of neferine in AD, PD, and other neurogenic diseases. The figure shows the neuroprotective potential of neferine in neurodegenerative diseases by modulating oxidative stress and neuroinflammation.

**Figure 4 biomolecules-15-01377-f004:**
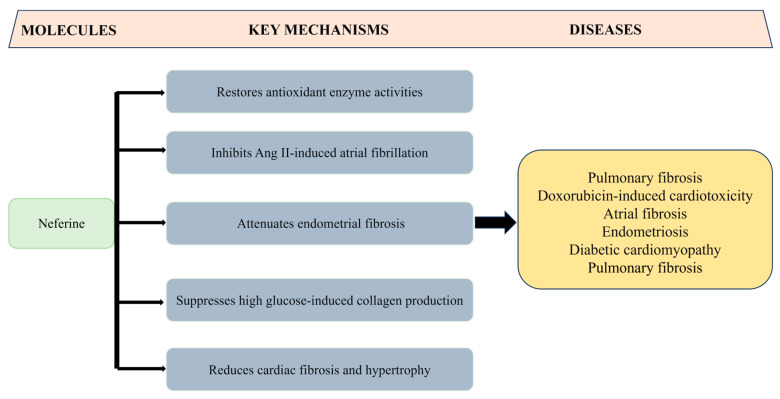
Proposed mechanisms underlying the antifibrotic actions of neferine. Highlights the antifibrotic activities of the alkaloid in different fibrosis models.

**Figure 5 biomolecules-15-01377-f005:**
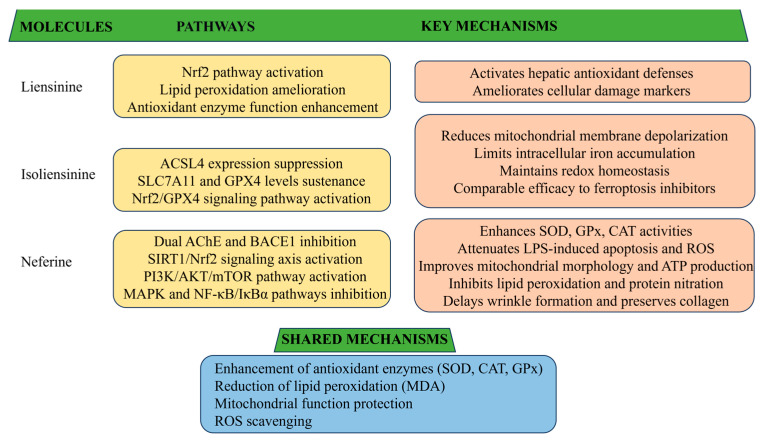
Proposed mechanisms underlying the antioxidant actions of liensinine, isoliensinine, and neferine, showing that how the alkaloids reduce oxidative stress and protect cells in various models.

**Figure 6 biomolecules-15-01377-f006:**
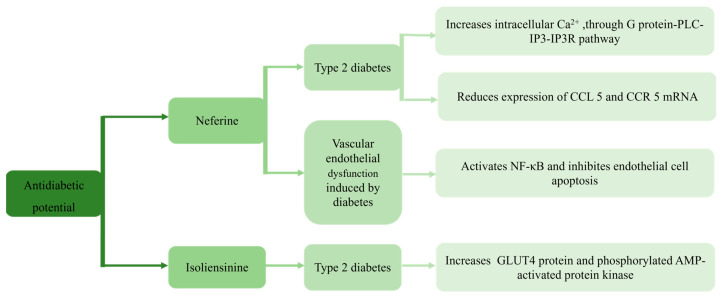
Proposed mechanisms underlying the antidiabetic actions of isoliensinine and neferine in the treatment of different diseases, showing the potential of isoliensinine and neferine on glucose uptake and vascular dysfunction in diabetic models.

**Table 1 biomolecules-15-01377-t001:** Mechanism of anti-inflammatory action of liensinine, isoliensinine, and neferine in the treatment of different diseases.

Alkaloid	Disease	Pathway
Liensinine	Acute liver injury	Inhibiting ferritinophagy and autophagosome-lysosome fusion
	Acute lung injury	Regulating PI3K/AKT/mTOR pathway, reducing LPS-induced acute lung injury and inflammatory factors
	Intestinal injury	Inhibiting NF-κB phosphorylation and NLRP3 inflammasome synthesis
	Chondrocyte inflammatory	Inhibiting NF-κB signaling pathway
Isoliensinine	Osteoarthritis	Inhibiting MAPK/NF-κB pathway activation, reducing extracellular matrix degradation, NLRP3, MMP 3 protein expression, and inflammation
Neferine	Atopic dermatitis-like skin Inflammation	Decreasing phosphorylation of p38, JNK, and ERK proteins
	Vascular inflammatory	Inhibiting NF-κB signaling, reducing mRNA and protein expression of ICAM1 and VCAM1
	Human lysophosphatidylcholine-stimulated human umbilical vein endothelial cells	Regulating DDAH-ADM pathway by restoring DDAH activity, increasing NO concentration, and decreasing ADMA, ROS, and MDA levels

**Table 2 biomolecules-15-01377-t002:** Mechanism of Antihypertensive action of liensinine, isoliensinine, and neferine in the treatment of different diseases.

Alkaloid	Disease	Pathway
Liensinine	Ang II-induced hypertension	Inhibits MAPK and TGF-β1/Smad2/3 signaling, reduces collagen deposition, and attenuates aortic wall thickening
Isoliensinine	Ang II- induced proliferation	Antagonizes Ang II effects, downregulates PDGF-β, bFGF, proto-oncogenes and c-Fos/c-Myc, and inhibits VSMC proliferation
Neferine	Pulmonary hypertension	Reduces RVSP, inhibits maladaptive right ventricular hypertrophy, restores extracellular matrix homeostasis, and suppresses proliferative signaling
	Hypertensive cardiomyocyte apoptosis	Inhibits activation of PI3K/AKT and TGF-β1/Smad2/3 signaling pathways
	Hypertensive vascular remodeling	Reduces cardiomyocyte apoptosis, restores mitochondrial membrane potential, and decreases ROS accumulation

## Data Availability

No data.

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
