# Peer review of "Pharmacological Potential and Mechanisms of Bisbenzylisoquinoline Alkaloids from Lotus Seed Embryos"

_biomolecules, 2025, doi:10.3390/biom15101377_

Round 1

Reviewer 1 Report

Comments and Suggestions for Authors

Another review on bisbenzylisoquinoline alkaloids from lotus nucifera. Similar reviews have been published in recent years (PMID: 34845950, 30404216, 34779018, 29499283). A major lack of originality and interest.

The focus on Lotus plumule is not justified. The 3 alkaloids can be found in other parts of the plant, they are not specific to the plumule (lotus embryos). Why this focus? Moreover, the extraction process and content should be discussed, at least briefly. The consumption of hot beverages (tea) prepared from plumule seeds or extracts will provide only tiny amounts of bisbenzylisoquinoline alkaloids. Quantitative data should be provided (alkaloid content per gram of plumule). Can a pharmacological effect be really expected when consuming teas from Lotus plumule? These aspects should be discussed.

The long anticancer section refers to the many studies performed with the alkaloids, essentially using in vitro systems (cell lines). In vivo data should be discussed separately, to highlight evidence of antitumor action in tumor-bearing animal models, if any. Moreover, a conclusion to the antitumor section (2.) is needed. It is the same with the other sections.

The authors seem to encourage the consumption of lotus plumule. They should also consider the associated risk. Lotus plumule also contains detrimental alkaloids, such as higenamine which is a prohibited product in sports. This point should be underlined in the discussion/conclusion (see PMID: 31973198, 35161335).

The abstract is too short, too general, and superficial; it lacks specific information. It should be extended to provide pharmacological information and a key message.

All Figures are the same type and shape. This is not very attractive. Variations would be preferred. An illustration of lotus plumule can be added. Some of the Figures (e.g. Fig. 3) are very superficial, not informative in the present form. Figures are not cited in the text.

The 2-pages list of abbreviations is excessive (useless). The format of references is non-homogeneous (pages missing in some cases).

A major revision is needed.

Comments on the Quality of English Language

There are errors of language ad typing errors in the text and Figures (e.g. Fig 2., antitumous).

Author Response

Another review on bisbenzylisoquinoline alkaloids from lotus nucifera. Similar reviews have been published in recent years (PMID: 34845950, 30404216, 34779018, 29499283). A major lack of originality and interest.

We greatly appreciate your review of our manuscript. In recent years, several reviews have been published, foucing on the bioactive compounds derived from lotus, highlighting the growing attention towards the important food and medicinal resource, which demonstrates broad potential for application. We have also reviewed numbers of papers and learned the related references that you mentioned, finding that alkaloids derived from lotus possess various bioactivities. Furthermore, we noted that bisbenzylisoquinoline alkaloids, including liensinine, isoliensinine, and neferine, are present in high concentrations in lotus seed embryos. Considering that these three bisbenzylisoquinoline alkaloids share a high degree of structural similarity, they likely exhibit similar bioactivities and mechanisms of action. Previous reviews have not systematically analyzed the functional similarities and diverse differences among these three classes. Through our review, we hope to provide a more comprehensive and in-depth understanding of the bioactivities and mechanisms of action of these bisbenzylisoquinoline alkaloids, which may offer more references for the utilization of bioactive compounds from lotus.

The focus on Lotus plumule is not justified. The 3 alkaloids can be found in other parts of the plant, they are not specific to the plumule (lotus embryos). Why this focus? Moreover, the extraction process and content should be discussed, at least briefly. The consumption of hot beverages (tea) prepared from plumule seeds or extracts will provide only tiny amounts of bisbenzylisoquinoline alkaloids. Quantitative data should be provided (alkaloid content per gram of plumule). Can a pharmacological effect be really expected when consuming teas from Lotus plumule? These aspects should be discussed.

We sincerely thank you for the insightful comment. The three alkaloids are not exclusive to the lotus plumule, but can be detected in other parts of Nelumbo nucifera, such as leaves, seed coats, and rhizomes.

The reasons that why we focus on lotus seed embryos rest on some considerations: Firstly, The lotus plumule is one of the richest tissues in terms of bisbenzylisoquinoline alkaloid content. Previous phytochemical studies have demonstrated that the concentrations of liensinine, isoliensinine, and neferine are significantly higher in the plumule compared to other parts of the plant, making it the most efficient source for extraction and pharmacological studies. Secondly, In traditional Chinese medicine, lotus plumule has been used for centuries as a medicinal material, particularly for its effects on cardiovascular, neurological, and inflammatory disorders. Its historical and clinical use highlights its value as a pharmacological resource. At the same time, Most of the cited pharmacological studies have employed purified alkaloids obtained from lotus plumule rather than other plant parts, ensuring consistency in experimental evaluation and supporting the rationale for focusing on this tissue. In light of these points, we believe that focusing on lotus plumule is meaningful.

As you suggested, we have added the overview of extraction methods and quantitative data in the revised version. We also stated that the actual dosage consumed when drinking lotus seed heart tea is significantly lower than that used in pharmacological experiments, and therefore should not be directly equated with the clinical efficacy observed in tea consumption.

We added the rationale for focusing on lotus seeds, extraction technique and quantitative data in the introduction, and the explanation of its limitation in the discussion, emphasizing that pharmacological effects require standardized extraction or preparation development rather than simply drinking them daily.

The long anticancer section refers to the many studies performed with the alkaloids, essentially using in vitro systems (cell lines). In vivo data should be discussed separately, to highlight evidence of antitumor action in tumor-bearing animal models, if any. Moreover, a conclusion to the antitumor section (2.) is needed. It is the same with the other sections.

We sincerely thank you for this valuable suggestion.

As you suggested, in the anti-cancer section, the paper has been restructured into two main parts. The first part details the mechanisms and discoveries of liensinine, isoliensinine, and neferine in vitro cell lines, providing a comprehensive overview of their multifaceted anticancer effects at the cellular level. The second part presents findings from in vivo animal models (such as xenograft models and chemically induced models), highlighting the translational potential and in vivo efficacy of these alkaloids, particularly liensinine and neferine.

We have also added the concluding paragraph to synthesize key findings. It summarizes the critical mechanisms shared by the three alkaloids, emphasizes the importance of emerging in vivo evidence, and outlines necessary future research steps.

Using the same approach, we appended short conclusion paragraphs to each main section (neuroprotective, anti-inflammatory, antihypertensive, antifibrotic, antiarrhythmic, antioxidant, antidiabetic, and other effects) that summarize principal evidence, mechanisms and translational limitations, and provide concise future directions.

The authors seem to encourage the consumption of lotus plumule. They should also consider the associated risk. Lotus plumule also contains detrimental alkaloids, such as higenamine which is a prohibited product in sports. This point should be underlined in the discussion/conclusion (see PMID: 31973198, 35161335).

We sincerely thank you for raising the concern regarding the anti-doping risk associated with lotus plumule. In this paper, we analyzed the bioactive potential of three major bisbenzylisoquinoline alkaloids, namely liensinine, isoliensinine, and neferine. As you suggested, we have now addressed the presence of these detrimental alkaloids in the discussion section to clarify that our review does not encourage the consumption of lotus plumule, and we hope to avoid any potential misinterpretation.

The abstract is too short, too general, and superficial; it lacks specific information. It should be extended to provide pharmacological information and a key message.

Thank you for your valuable feedback regarding the abstract of our manuscript. As you suggested, we have revised the abstract to provide a more comprehensive and detailed summary of our review. 

All Figures are the same type and shape. This is not very attractive. Variations would be preferred. An illustration of lotus plumule can be added. Some of the Figures (e.g. Fig. 3) are very superficial, not informative in the present form. Figures are not cited in the text.

We sincerely thank you for these critical and constructive suggestions regarding the figures. As you suggested, we have undertaken a comprehensive revision of all figures, as detailed below.

A detailed anatomical illustration of the lotus plumule has been added as Figure 1 in the Introduction section. This figure visually highlights the morphological structure.

The original superficial figures (Fig. 3 on Antihypertensive effects) have been completely redesigned and replaced. The new table are now detailed, mechanistic schematics that accurately depict the specific molecular pathways and targets through which each alkaloid exerts its pharmacological effects for a given disease. They are now data-rich and informative.

 As you suggested, we have cited the figures in the paper.

The 2-pages list of abbreviations is excessive (useless). The format of references is non-homogeneous (pages missing in some cases).

As you suggested, we have removed the 2-pages list of abbreviations and all abbreviations were defined when they first appeared in the text. We have thoroughly checked and unified the formatting of all references.

A major revision is needed.

We fully acknowledge the reviewer's assessment that a major revision was necessary for our initial submission. We sincerely apologize for the numerous shortcomings in the original manuscript, which fell short of the expected standards for clarity, organization, and scholarly presentation.

In response, we have undertaken a comprehensive and meticulous revision of the entire manuscript to address every point raised by all the reviewers. 

We have clarified the unique focus and novelty of our review in the Introduction, explicitly distinguishing it from previously published works.

Following the reviewer's recommendations, we have substantially restructured the manuscript. This includes dividing the lengthy "Antitumor effects" section into distinct sections: in vitro research and in vivo research, while adding an analytical summary . We have also streamlined other sections to ensure coherent and logically structured content.

The manuscript has been edited and underwent multiple rounds of proofreading to eliminate all grammatical, typographical, and formatting errors. This includes correcting the error in Figure 2.

Figure 3 have been redesigned . A new illustration of the lotus plumule has been added as Figure 1.

The list of references has been checked for consistency and formatting, with all missing elements added and is now fully uniform.

Reviewer 2 Report

Comments and Suggestions for Authors

The purpose of this bibliographic study is to synthesize and systematize the current knowledge available in the scientific literature regarding the bioactive potential of alkaloids identified in lotus, as well as to assess their possible therapeutic applications in clinical practice.The evidence discussed in this manuscript is suggestive and highlights the potential applicability of these natural products in the prevention and treatment of a range of pathological conditions.I consider that the word effect should be replaced with the word potential when describing possible bioactive effects. In addition, you should also address the possibilities of obtaining these alkaloids from lotus, particularly isoliensinine, liensinine, and neferine, for which the description is more extensive. In the cited articles, was their bioactive potential evaluated on these isolated alkaloids obtained from total extracts, or was it assessed on the crude extract as a whole and subsequently correlated with the presence of these alkaloids?

Assuming that aspects of technological transfer have already been discussed, what would be the most appropriate framework for developing a therapeutic regimen?

Incorporating these data would increase the depth and complexity of the manuscript while also opening new avenues for future research in this direction!

Author Response

The purpose of this bibliographic study is to synthesize and systematize the current knowledge available in the scientific literature regarding the bioactive potential of alkaloids identified in lotus, as well as to assess their possible therapeutic applications in clinical practice.The evidence discussed in this manuscript is suggestive and highlights the potential applicability of these natural products in the prevention and treatment of a range of pathological conditions.I consider that the word effect should be replaced with the word potential when describing possible bioactive effects. In addition, you should also address the possibilities of obtaining these alkaloids from lotus, particularly isoliensinine, liensinine, and neferine, for which the description is more extensive. In the cited articles, was their bioactive potential evaluated on these isolated alkaloids obtained from total extracts, or was it assessed on the crude extract as a whole and subsequently correlated with the presence of these alkaloids?

Thank you for your valuable feedback. We have addressed your comments as follows.

We appreciate your suggestion regarding the use of the term "effect" and have updated the manuscript accordingly. We replaced "effect" with "potential" in each of the subheadings, especially when discussing the bioactive effects of alkaloids. This change better reflects the scope of the research and emphasizes the promising, yet not fully established, therapeutic properties of these compounds.

In response to your query regarding the extraction of alkaloids from lotus, particularly isoliensinine, liensinine, and neferine, we have expanded the discussion. We clarified that the alkaloids are primarily isolated from lotus seed embryos using pH-zone-refining countercurrent chromatography, which provides a high yield of these compounds from germinated lotus seed embryos. This detail is now included to offer more insight into the extraction process.

Regarding your question on whether the bioactive potential of these alkaloids was assessed from isolated alkaloids or crude extracts, the studies cited in the manuscript evaluated the bioactive potential of the alkaloids themselves (liensinine, isoliensinine, and neferine) rather than using crude extracts. We specifically referenced studies that assessed these alkaloids in isolated forms to establish their therapeutic potential, ensuring that the contributions of these compounds are properly attributed to their individual actions.

Assuming that aspects of technological transfer have already been discussed, what would be the most appropriate framework for developing a therapeutic regimen?

We sincerely thank the reviewer for this important suggestion. In the revised manuscript, we have added a concise discussion of the potential framework for therapeutic development of bisbenzylisoquinoline alkaloids from lotus seed embryos. Based on current preclinical data, we propose a multi-step framework: mechanistic validation to confirm key signaling pathways across different disease models; dose–response studies to establish therapeutic windows, pharmacokinetics, and pharmacodynamics; formulation optimization to improve solubility and bioavailability; safety and toxicity assessment in animal models; and translation to early-phase clinical trials with clearly defined biomarkers for efficacy and safety. This staged framework is consistent with general drug development principles and highlights the need for rigorous validation before clinical application. We have integrated this point into the discussion section to strengthen translational relevance.

Incorporating these data would increase the depth and complexity of the manuscript while also opening new avenues for future research in this direction!

Thank you very much for this positive feedback and for recognizing the potential of our work to deepen the scientific understanding of lotus alkaloids and inspire future research directions. We greatly appreciate your encouraging comments.

Reviewer 3 Report

Comments and Suggestions for Authors

The authors of the review "Pharmacological activities of bisbenzylisoquinoline alkaloids derived from lotus seed embryos" describe in sufficient detail the data of pharmacological studies of three closely related alkaloids of this plant. The emphasis is not on the pharmacological properties themselves, but on the molecular mechanisms of each type of biological activity. The listing of individual markers of the effect of alkaloids on cell cultures or animal models looks cumbersome and difficult to read. Although the authors made summary tables and figures, the text to them within the paragraph is still poorly systematized. Very long paragraphs that can be divided into separate semantic subsections.

As an example, I will analyze the paragraph on antitumor activity:

In the first paragraph, chemotherapy is contrasted with therapy with compounds of natural origin. Although there are many biomolecules among chemotherapy drugs.

In the same paragraph, the activity of neferine is first discussed, with an emphasis on combined use with known antitumor drugs. Then two other alkaloids, and then neferine again. Moreover, with a complete repetition of the text from the beginning of the paragraph. It turns out to be chaotic. This paragraph lists numerous markers, but there is no analysis of the information on what is common to the three alkaloids, and how they differ.

There is no information on doses and concentrations at all, there is no way to evaluate the selectivity of the antitumor effect.

In general, the review text is very sloppy: repetitions, poor formatting, different fonts, lost citations, etc.

There is no information on how specific these alkaloids are. But since the authors state in the conclusion that only lotus buds are a compelling candidate for future drug development, then apparently they are not found anywhere else.

I would also like to see an analysis of the information on whether the diverse effects are a consequence of action on the same targets or whether the compounds are extremely non-selective and act unpredictably on different biological functions. Or do different effects occur at different dosages?

And please shorten the texts.

Author Response

The authors of the review "Pharmacological activities of bisbenzylisoquinoline alkaloids derived from lotus seed embryos" describe in sufficient detail the data of pharmacological studies of three closely related alkaloids of this plant. The emphasis is not on the pharmacological properties themselves, but on the molecular mechanisms of each type of biological activity. The listing of individual markers of the effect of alkaloids on cell cultures or animal models looks cumbersome and difficult to read. Although the authors made summary tables and figures, the text to them within the paragraph is still poorly systematized. Very long paragraphs that can be divided into separate semantic subsections.

As an example, I will analyze the paragraph on antitumor activity:

In the first paragraph, chemotherapy is contrasted with therapy with compounds of natural origin. Although there are many biomolecules among chemotherapy drugs.

In the same paragraph, the activity of neferine is first discussed, with an emphasis on combined use with known antitumor drugs. Then two other alkaloids, and then neferine again. Moreover, with a complete repetition of the text from the beginning of the paragraph. It turns out to be chaotic. This paragraph lists numerous markers, but there is no analysis of the information on what is common to the three alkaloids, and how they differ.

There is no information on doses and concentrations at all, there is no way to evaluate the selectivity of the antitumor effect.

We sincerely thank you for your detailed and constructive comments on our review. This article focuses more on the main mechanisms of action in various diseases than on its own pharmacological activity, so we changed the title to Pharmacological Potential and Mechanisms of Bisbenzylisoquinoline Alkaloids from Lotus Seed Embryos. The title is more relevant to the actual content of the article.

We have restructured the sections to enhance clarity and readability. Long paragraphs have been broken down into smaller, more digestible subsections, each focusing on a distinct aspect of the pharmacological effects of bisbenzylisoquinoline alkaloids. This restructuring ensures that the key concepts are presented more clearly and logically. As for the repetition of text content and the disorder of text in the first paragraph you mentioned, we have changed the content, deleted most of the content. The antitumor section has been reorganized into in vitro evidence and In vivo evidence, each followed by a concise conclusion paragraph. Within the in vitro subsection, the activities of each alkaloid are now discussed in distinct, sequential paragraphs: first liensinine, then isoliensinine, and finally neferine. This restructuring avoids excessively long paragraphs and improves logical flow and readability. Redundant descriptions of neferine have been deleted, and the sequence of presentation has been adjusted. Numerous molecular markers and signaling details have been consolidated into tables and a new mechanistic figure, with the main text restricted to integrative summaries. We have also streamlined the discussion of individual markers and mechanisms, focusing on the most relevant ones for each biological activity. The detailed analysis of the molecular mechanisms remains intact, but we have reduced excessive detail

We have now integrated the tables and figures into the text, referring to them where necessary to illustrate key points. This should improve the flow of information and reduce redundancy, making the manuscript easier to read and follow.

The revised text highlights both shared mechanisms (e.g. modulation of PI3K/AKT, MAPK/JNK, NF-κB) and unique features (e.g. liensinine’s inhibition of Kv10.1, isoliensinine’s AKT-binding, and neferine’s ROS–MAPK activation).

In the modified article, several concentrations of alkaloids have been added to understand their scope and selectivity more intuitively.

We have also added a concluding noting that synthesizes the information. This conclusion explicitly analyzes the commonalities and differences between the three alkaloids and added a discussion noting that most effective concentrations in vitro are in the micromolar range, whereas plasma concentrations achievable from traditional tea preparations are typically nanomolar—underscoring a significant translational gap. We also point out that the pleiotropic effects may in part reflect dose dependence and limited selectivity, emphasizing the need for future pharmacokinetic and standardized formulation studies.

In general, the review text is very sloppy: repetitions, poor formatting, different fonts, lost citations, etc.

We sincerely and deeply apologize for the overall sloppiness of our initial submission, which fell below the expected standards of academic rigor and presentation. We fully acknowledge the criticisms regarding repetitions, poor formatting, inconsistent fonts, and lost citations. We have undertaken a thorough, line-by-line revision of the entire manuscript to comprehensively address these issues.

There is no information on how specific these alkaloids are. But since the authors state in the conclusion that only lotus buds are a compelling candidate for future drug development, then apparently they are not found anywhere else.

We sincerely thank the reviewer for raising this important point. The three bisbenzylisoquinoline alkaloids liensinine, isoliensinine, and neferine are not exclusively found in lotus plumule but have also been detected in other parts of the lotus plant, including leaves and seeds. Our decision to focus on lotus plumule is based on its long-standing use in traditional medicine and dietary applications, as well as the relatively higher content of these alkaloids reported in this tissue compared with other plant parts.

Regarding specificity, current evidence suggests that these alkaloids are not highly selective but act on multiple signaling pathways. Their diverse pharmacological effects may in part reflect this multi-target nature, as well as dose-dependent responses.

We have added a discussion in the revised manuscript emphasizing that while lotus plumule is an abundant and practical source of these compounds, the compounds themselves are not unique to this tissue. Furthermore, we highlight the need for systematic pharmacological and pharmacokinetic studies to clarify their selectivity and the translational implications of their dose-dependent actions. The expression that only lotus heart buds are a compelling candidate for future drug development has been removed from the discussion.

I would also like to see an analysis of the information on whether the diverse effects are a consequence of action on the same targets or whether the compounds are extremely non-selective and act unpredictably on different biological functions. Or do different effects occur at different dosages?

We sincerely thank the reviewer for this insightful comment.

Based on the available evidence, liensinine, isoliensinine, and neferine are not highly selective agents acting on a single molecular target, but rather exhibit multi-target pharmacological activity. Several of their diverse effects converge on common signaling pathways such as PI3K/AKT, MAPK/JNK, NF-κB, and TGF-β/Smad, which underlie their shared antioxidant, anti-inflammatory, antitumor, and neuroprotective properties. At the same time, each compound also demonstrates distinct interactions—for example, liensinine modulates Kv10.1 channels and FGFR2 G-quadruplexes, isoliensinine binds directly to AKT and TGFBR1, while neferine uniquely induces ROS–MAPK–mediated pyroptosis and reverses chemoresistance.

Importantly, many of these effects appear to be dose-dependent. In vitro studies consistently report effective concentrations in the micromolar range, while lower concentrations often yield weaker or different biological responses. For example, neferine exhibits greater potency than liensinine in suppressing arrhythmias at sub-10 μM levels, whereas liensinine’s cytotoxic effects in osteosarcoma are most pronounced at ~80 μM. These findings suggest that the apparent pleiotropy is partly explained by differential pathway sensitivity to concentration.

We have incorporated this analysis into the revised manuscript, emphasizing that the multifaceted actions of these alkaloids likely reflect both multi-target pharmacology and dosage-dependent mechanisms. However, the possibility of non-specific or off-target effects cannot be excluded, underscoring the need for systematic pharmacological and pharmacokinetic studies to define target selectivity, therapeutic windows, and translational potential.

And please shorten the texts.

We sincerely thank the reviewer for this valuable suggestion. We carefully revised the manuscript and substantially shortened the text throughout. Repetitive descriptions were removed, long paragraphs were divided into shorter sections, and only the most representative experimental findings and signaling pathways were retained. For example, in the antitumor and neuroprotection sections, we reorganized the content to emphasize shared mechanisms and distinctive features rather than listing all markers. Likewise, in other pharmacological sections, redundant details were condensed, and key mechanisms were summarized in tables and figures for clarity.

Round 2

Reviewer 1 Report

Comments and Suggestions for Authors

Most of my previous comments have been considered and appropriate changes have been made. The revised manuscript is improved and may be published, pending a minor revision.

Fig 2 is still very scholar and trivial. The term “anti-pressure” is not appropriate. There is a mix of pathology-related terms (antitumor, antidiabetic…) and property-related terms (antioxidant, anti-inflammatory, …). This Fig should be redrawn correctly or simply removed.

Author Response

Most of my previous comments have been considered and appropriate changes have been made. The revised manuscript is improved and may be published, pending a minor revision.

Fig 2 is still very scholar and trivial. The term “anti-pressure” is not appropriate. There is a mix of pathology-related terms (antitumor, antidiabetic…) and property-related terms (antioxidant, anti-inflammatory, …). This Fig should be redrawn correctly or simply removed.

Thank you very much. As you suggested, we have removed Fig2.